# Relationships between Perfectionism, Extra Training and Academic Performance in Chinese Collegiate Athletes: Mediating Role of Achievement Motivation

**DOI:** 10.3390/ijerph191710764

**Published:** 2022-08-29

**Authors:** Chengjiang Han, Feng Li, Bizhen Lian, Tomas Vencúrik, Wei Liang

**Affiliations:** 1China Basketball College, Beijing Sport University, Beijing 100084, China; 2Department of Sports, Faculty of Sports Studies, Masaryk University, 62500 Brno, Czech Republic; 3Centre for Health and Exercise Science Research, Hong Kong Baptist University, Hong Kong 999077, China

**Keywords:** personality, achievement motivation, perfectionism, collegiate athlete, basketball players, mediation, extra training, education

## Abstract

There are limited studies examining the impacts of perfectionism and achievement motivation on collegiate athletes’ extra training and academic achievement in a Chinese context. This study aimed to examine the association of perfectionism (five facets) with extra training and academic performance among Chinese collegiate athletes and identify the mediating role of achievement motivation (two attributes) in the relationship between perfectionism and extra training and academic performance. With a prospective study design, 243 eligible participants completed two-wave surveys from September to December 2021. Measures included demographics, perfectionism (concern over mistake, CM; doubts about action, DA; personal standard, PS; organization; parental expectation, PE), achievement motivation (motive for success, MS; motive for avoiding failure, MF), extra-training (minutes/week), and academic performance (GPA). Results showed that CM, DA, PS, and MS were associated with extra training among Chinese collegiate athletes, while the associations of DA and PS with extra training were mediated by MS. In addition, DA, PS, organization, and MS were associated with participants’ GPA, while MS was a salient mediator for the contributions of DA and PS on participants GPA. Research findings give new insights to the psychological mechanisms of perfectionism and achievement motivation on collegiate athletes’ extra training and academic performance, contributing to future studies in relevant domains.

## 1. Introduction

Personality, a core psychological characteristic of an individual, reflects the consistent and stable patterns of an individual’s thoughts, feelings, and behaviors [1,2]. Different personalities show different behavioral characteristics in the pursuit of personal goals [2]. Perfectionism, a personality trait, is characterized by an individual’s concerns with striving for flawlessness and perfection and is accompanied by critical self-evaluations and concerns regarding others’ evaluations [3,4]. People who pursue high perfection in their own performance, set high requirements/standards for themselves, and emphasize that everything they undertake should be organized, are known as “perfectionists” [3,4]. Traditionally, perfectionism was regarded as a single-dimensional personality disposition which is a sign of psychological maladjustment and disorders as individuals seeking psychological support for depression and anxiety often show elevated levels of perfectionism [5,6]. However, with the development of theories and empirical evidence in the last 20 years, a new consensus has emerged that conceptualized perfectionism as a multidimensional and multifaceted trait, which includes two higher-order dimensions: *perfectionistic concerns* and *perfectionistic strivings* [5,6,7,8]. Perfectionistic concerns capture several facets of perfectionism associated with concern over mistakes, feelings of discrepancy between one’s expectations and performance, and negative reactions to imperfection (doubts about action), while perfectionistic strivings capture those aspects associated with self-oriented striving for perfection and setting exceedingly high standards of performance [5,6]. Perfectionism has been demonstrated with a series of social-cognitive and behavioral outcomes (e.g., motivation, sport, education) [6,7]. Research on the perfectionism has also extended from emphasizing negative characteristics and only behavioral aspects to underlining positive aspects and in-depth exploration of multiple structural characteristics [5]. Most existing studies focused on the two higher-order dimensions of perfectionism, which found that perfectionistic concerns show consistent positive relationships with maladaptive outcomes (e.g., burnout), while perfectionistic strivings show positive relationships with both maladaptive and adaptive outcomes (e.g., academic engagement and good performance) [7,8,9,10]. However, research on the specific aspects within each dimension (e.g., concern over mistake and doubts about action for the perfectionistic concerns dimension, and personal standard, organization, and parental expectation for the perfectionistic strivings dimension) is limited. In this regard, differentiating these specific facets of perfectionism is crucial as they show distinct patterns of relationships with various outcomes.

In the sports domain, perfectionism is considered a crucial personality trait, which is regarded as a common characteristic for world Olympic champions [11]. Perfectionism is also closely related to competition anxiety, achievement motivation, psychological fatigue, sports performance, and other sports-related activities [12,13,14,15]. Hardy et al. stated that athletes who excessively pursue perfection tend to set higher personal standard for themselves and show better organization, which can obtain satisfaction from their own behavior and produce lower pressure [16]. By contrast, for athletes who pursue perfection, higher expectations from the outside may result in psychological pressure, which has a negative influence on athletes’ performance [17,18]. In addition, athletes with perfectionism not only tend to complete the training tasks assigned by the coach, but also conduct extra training to improve themselves, whereby they improve their performance in the competition [17]. Extra training can help automate athletes’ skills, reducing the negative impact of psychological stress on sport performance [17]. According to the literature, studies related to extra training mainly focus on the medical field to improve patients’ physical functions [19,20], while research in the sports domain is still limited. Two studies indicated a positive association of perfectionistic strivings with engagement in sport-specific activities (including extra training) in team sport athletes in Western countries, but these studies did not particularly focus on extra training, and the relationship between perfectionistic concerns and behavioral outcomes is still unclear [8,21]. There is also a scarcity of evidence examining this issue in a Chinese context.

Perfectionism can not only affect individuals’ psychological fatigue and sports performance, but also promote the improvement of academic performance. Studies showed that students with perfectionism generally present outstanding qualifications and excellent performance [22,23]. They further pointed out that the proportion of the number of students becomes more and more obvious with the increase of age and grade. Previous study reported that students with perfectionism emphasize personal standard and organization, showing higher motivation for success [24]. Therefore, they tend to devote more efforts to achieve their ideals and goals. Previous studies regarding perfectionism and academic performance mainly targeted middle school students or collegiate students, while studies with collegiate athletes are still limited.

In addition, the mediation mechanism of the impact of perfectionism on extra training and academic performance has raised increasing concerns recently, where the achievement motivation has been suggested as a potential mediator [25]. Generally, the motivation is to inspire and maintain the organism’s action, and action leads to a goal psychological tendency or internal drive [25,26], which is an important source of motivation for its success. Achievement motivation refers to the internal motivation for people to succeed in the process of completing tasks, as well as an internal driving force that people are willing to do what they believe is important and valuable and strive for perfection [27]. Many studies showed that achievement motivation is mainly manifested in the *motive for success* and *motive for avoiding failure* [28,29]. In addition, it is an impulse of competition between internal and external standards of excellence, with differences mainly reflected in individual social orientation and ego orientation [30]. Some studies indicated that motive for success is positively associated with performance while motive for avoiding failure is negatively related to performance [31,32,33]. However, no relationship between achievement motivation and academic achievement was also found in other studies [33]. Given the mixed findings, further examination on this relationship is needed, especially in the special samples of collegiate athletes as they need not only spend time on sports training, but also on their studies. In addition, achievement motivation has proven to be a mediator between perfectionism and other outcomes (e.g., subjective well-being) [34], while to the best of our knowledge, there are limited studies examining whether the achievement motivation is the intermediary variable between perfectionism and extra training and academic performance.

Given the above, the current study aimed to explore the association of five facets of perfectionism with extra training and academic performance among Chinese collegiate athletes, and to identify whether the two attributes of achievement motivation play a mediating role in the association of perfectionism with extra training and academic performance. From a theoretical perspective, both perfectionistic concerns and perfectionistic strivings could be associated with extra training and academic achievement [6,7], particularly when individuals have overly critical evaluations and concerns about making mistakes and doubt their action. These perfectionistic concerns are associated with worry, rumination, and other maladaptive cognitions that may stifle productive behavior, where individuals with high perfectionistic concerns may experience overwhelming feelings of pressure [7]. Consequently, they may be more concerned about avoiding mistakes and spend less time on relevant activities and instead procrastinate as a method to avoid facing potential failure. By contrast, individuals with high perfectionistic strivings often set exceptionally high standards that will direct, motivate, and regulate behaviors that are beneficial for a better performance. Individuals may spend more time on relevant activities to achieve success.

We, therefore, hypothesized that:(a)the aspects of perfectionistic concerns (i.e., concern over mistake and doubts about action) would show direct and negative association with extra training (hypothesis 1a);(b)the aspects of perfectionistic strivings (i.e., personal standard, organization and parental expectation) would show direct and positive association with extra training (hypothesis 1b);(c)two attributes of achievement motivation (i.e., motive for success and motive for avoiding failure) would mediate the relationship between perfectionistic concerns aspects and extra training (i.e., the direct effect would be attenuated; hypothesis 1c)(d)the aspects of perfectionistic concerns (i.e., concern over mistake and doubts about action) would show direct and negative association with academic performance (GPA) (hypothesis 2a);(e)the aspects of perfectionistic strivings (i.e., personal standard, organization and parental expectation) would show direct and positive association with academic performance (GPA) (hypothesis 2b);(f)two attributes of achievement motivation (i.e., motive for success and motive for avoiding failure) would mediate the relationship between perfectionistic concerns aspects and academic performance (GPA) (i.e., the direct effect would be attenuated; hypothesis 2c). The research framework is shown in Figure 1.

## 2. Materials and Methods

### 2.1. Study Design, Participants, and Procedure

The study used a prospective design with two-wave data collection. We contacted 260 collegiate basketball players from colleges and universities in the cities of Beijing, Wuhan, Shanxi, and Fujian, using a convenience sampling method. All eligible participants met certain criteria, including (1) aged ≥ 18 years; (2) being a member of representative sport team of university or having more than 3 years basketball training background; and (3) sufficient language skills in Chinese. The paper questionnaires were distributed to the participants before and after the regular training sessions with the assistance of coaches. A total of 255 eligible participants completed the first-wave data collection during the last week of September 2021. Finally, 243 (203 boys, 40 girls, age: 20.56 ± 1.51 years) participants completed the second-wave data collection at the 3-month follow-up in December 2021. Based on the common rule of thumb for estimating the sample size for a structural equation modeling (SEM) (i.e., the sample size is usually 10 times the number of variables) [35,36], the sample size of 243 was sufficient for this study. All participants were informed about the study purpose and signed the written informed consent form before the data collection. Participants were asked to complete the paper questionnaires anonymously and independently. The data collection was conducted at the training venues and each survey lasted 15–20 min.

### 2.2. Measures

#### 2.2.1. First-Wave Data Collection

Demographics and perfectionism were measured in the first-wave data collection.

Demographic variables included age, gender, grade (freshman/sophomore/junior/senior), major (arts-related/sciences-related), and training duration.

For the Perfectionism Scale, the Chinese Version of Frost Multidimensional Perfectionism Scale (MPS-F) by Zi Fei (2006) [37] was adopted. In this study, five dimensions of the scale were applied: Concern over mistake (9 items, for example, “if I made a mistake, I will upset”), personal standard (7 items, such as “if I don’t set the highest standards for myself, I could be a mediocre person”), doubt about action (4 items, such as “even if I do one thing very carefully, also often think of things is not good enough”), organization (6 items, e.g., “Organization/organization is very important to me”), and parental expectation (5 items, e.g., “My parents set high standards for me”). The dimensions of personal standard and organization constitute well-adapted perfectionism, while the other three dimensions constitute maladaptive perfectionism. A total of 31 items were included in the scale, and a 5-point Likert score was used to mark “very inconsistent” to “very consistent” as 1–5 points. The scale is all positive scoring questions. The higher the total score and the higher the score of each dimension, the stronger the perfectionism tendency. Both subscales demonstrated acceptable internal reliability with Cronbach’s α = 0.863, Cronbach’s α = 0.702, Cronbach’s α = 0.739, Cronbach’s α = 0.725, and Cronbach’s α = 0.782, respectively.

#### 2.2.2. Second-Wave Data Collection

The achievement motivation, extra training time, and academic performance were measured at the second-wave data collection.

For Achievement Motivation Scale, Ye Renmin and Hagtvet’s version was used [38]. This scale consists of 30 items, which are divided into 2 subscales: *motive for success dimension* and *motive for avoiding failure*. The responses were given on a 4-point Likert scale, ranging from “4 = completely right” to “1 = completely wrong”. A higher total score of the scale (i.e., the score of the *motivation for success* subscale minus the score of the *motivation for avoiding failure* subscale) indicates a stronger achievement motivation. Both subscales demonstrated acceptable internal reliability with Cronbach’s α = 0.884 and Cronbach’s α = 0.873, respectively.

The extra training time (hours/week) was self-reported by the participants and the academic performance was measured by the semester grade point average (GPA). The semester GPA was collected directly from the university records, which was calculated using the credit-weighted sum of the grades for all courses divided by the total credits. The GPA was coded on a continuous scale ranging from “A = 4” to “F = 0 (failed)”.

### 2.3. Statistical Analysis

Data screening and primary analyses were conducted using the IBM SPSS 27.0 (Armonk, NY, USA). Invalid and abnormal data were cleaned prior to the data analyses. Data distribution was detected by the Q-Q plot and S-K test. Mean values, standard deviation, and percentage were used to present the descriptive information of study samples. Zero-order correlation between target variables was examined using the Pearson/Spearman correlation coefficients. Mplus 8.0 (Los Angeles, CA, USA) was employed for the mediation analyses. The model fit was evaluated using several goodness-of-fit indices, including Chi-square (*χ*^2^), Chi-squared/deviation freedom (*χ*^2^/*df*), comparative fit index (CFI), Tucker–Lewis fit index (TLI), root mean square error of approximation (RMSEA), and standardized root mean square residual (SRMR). The general criteria for an acceptable model fit were <5 for *χ*^2^/*df*, >0.90 for CFI and TLI, and <0.08 for RMSEA and SRMR [39,40]. For all direct and indirect effects in the path analysis, standardized coefficients (*β*) with 95% confidence intervals (CI) were calculated using maximum likelihood estimation with a bias-corrected bootstrapped approach (5000 resamples). All the demographics were included as covariates in the model analyses. All significance levels were set as *p* < 0.05 (two-tailed). The effect size of Cohen’s *f*^2^ for the model prediction was calculated by using the equation “*f*^2^ = *R*^2^/(1 − *R*^2^)”, with 0.02, 0.15, and 0.35 indicating a small, moderate, and large effect, respectively [41,42].

## 3. Results

### 3.1. Sample Characteristics and Primary Analysis

A total of 243 (83.5% males) collegiate basketball players were included in the analyses, with an average age of 20.56 years (SD = 1.51; 18–27 years). Most of the participants (65.8%) majored in science-related subjects. The percentage of freshman, sophomore, junior, and senior was 25.1%, 20.9%, 23.0%, and 21.0%, respectively. The average training duration of participants was 3.91 years (SD = 1.08; 1–5 years).

For the primary analysis, only a few of the scale items departed from the normality distribution and the absolute values of skewness and kurtosis were 1.12–1.47 (>1). The robust maximum likelihood estimation approach was therefore used in the CFA whereby the standard errors and tests of model fit were robust with respect to the observed variables with non-normal distribution [39,43]. As presented in Table 1, the measurement scales showed acceptable internal consistency reliability, with Cronbach’s *α* coefficients ranging from 0.70 to 0.88. The model-fit indices from seven preliminary CFAs of these scales indicated an acceptable-to-good fit to the data (χ^2^/*df* ≤ 2.39, CFI ≥ 0.94, TLI ≥ 0.92, RMSEA ≤ 0.08, and SRMR ≤ 0.05), with all item-factor loadings being acceptable (≥0.38). Finally, the inter-factor correlations did not encompass unity, indicating the distinction of concept among these factors.

Table 2 shows the Pearson correlation coefficients of study variables. Small-to-moderate correlations (*r* = 0.15–0.49) were found among these variables, indicating that there was no serious multicollinearity in the hypothesized mediation model.

### 3.2. Main Analysis

Both mediation models showed a satisfactory model-fit, with χ^2^/*df* = 1.39, CFI = 0.99, TLI = 0.95, RMSEA = 0.04, and SRMR = 0.03 for extra training, and χ^2^/*df* = 1.39, CFI = 0.99, TLI = 0.96, RMSEA = 0.04, and SRMR = 0.03 for academic performance. The overall model explained 22% and 37% of the variance in extra training (Cohen’s *f*^2^ = 0.28) and academic performance (Cohen’s *f*^2^ = 0.59), respectively.

As shown in Figure 2, for the direct effects of perfectionistic concerns aspects on extra training, results revealed a significant negative association of concern over mistake (*β* = −0.13, SE = 0.06, *p* = 0.021), and a positive association of doubts about action (*β* = 0.18, SE = 0.06, *p* = 0.005) with extra training. For the aspects of perfectionistic strivings, only personal standard (*β* = 0.13, SE = 0.06, *p* = 0.045) showed a significant correlation with extra training, while neither organization nor parental expectation was significantly associated with extra training (both *p* > 0.10). For two achievement motivation attributes, a significant association of extra training was only found on motive for success (*β* = 0.34, SE = 0.06, *p* < 0.001).

For the indirect effects, as shown in Table 3, two significant mediating paths were identified. Particularly, motive for success significantly mediated the association between personal standard and extra training (*β* = 0.10, *p* = 0.002), and between doubts about action and extra training (*β* = 0.06, *p* = 0.027).

For academic performance (GPA), 4 of 7 direct paths were found to be significant (Figure 3). Specifically, one aspect of perfectionistic concerns and two aspects of perfectionistic strivings showed significant associations with academic performance, including doubts about action (*β* = 0.12, SE = 0.06, *p* = 0.037), personal standard (*β* = 0.20, SE = 0.06, *p* = 0.001), and organization (*β* = 0.20, SE = 0.06, *p* < 0.001). For achievement motivation attributes, only motive for success was found to be significantly associated with participants’ academic performance (*β* = 0.33, SE = 0.06, *p* < 0.001).

For the indirect effects, motive for success was identified as a significant mediator for the associations of personal standard (*β* = 0.10, *p* =0.003) and doubts about action (*β* = 0.06, *p* = 0.037) with academic performance (see Table 3).

## 4. Discussion

This study explored the psychological mechanisms affecting extra training and academic performance of Chinese collegiate athletes from the perspectives of perfectionism and achievement motivation. In this study, we identified the direct effect of certain perfectionism aspects on extra training and academic performance in collegiate athletes. In addition, our findings also provided evidence for the mediating effects of the achievement motivation (i.e., motive for success) in the association of perfectionism with extra training and academic performance. Overall, the proposed mediation model showed a good fit, and study hypotheses were partially supported.

For extra training, both aspects of perfectionistic concerns (i.e., concern over mistake and doubts about action) were significantly associated with and accounted for a significant portion of the variance of extra training. Interestingly, these two associations showed different directions. For concern over mistake, it showed a negative correlation with extra training. Collegiate athletes with lower concern over mistake have a stronger tendency to extra training than those with higher. This result is consistent with a previous study reporting that athletes reducing their negative reactions to their own wrong movements improve their confidence and spend more time on training [44]. For doubts about action, it showed a positive association with extra training, which is contrary to our previous hypothesized association direction. As there is a lack of relevant evidence on examining the relationship between perfectionism and extra training, it is difficult to make a comparison. A previous study showed that athletes with higher doubts about their ability to complete tasks need more extra training to achieve their goals [45]. This may, to some extent, provide a potential explanation. However, more research on this point is needed. For the aspects of perfectionistic strivings, only personal standard showed a significant positive association with extra training. Collegiate students who had higher personal standard spent more time on extra training. This is consistent with previous suggestions and evidence [8,21]. It is noteworthy that a recent meta-analysis study concluded that perfectionistic strivings have a small-to-medium effect on a better performance in sport, while perfectionistic concerns were found to not be associated with performance [46]. Taken together with our findings, it suggests that different aspects of perfectionism may differ in the relationship with extra training among collegiate athletes and these relationships are complex and ambiguous. More research on examining these relationships is warranted.

In addition, personal standard and doubts about action have a significant effect on motive for success. Motive for success partially mediated the relationship between personal standard, doubts about action, and extra training. The result showed that there is a positive correlation between personal standard and achievement motivation, which explains a large part of the variance of achievement motivation. Previous studies reported that the pursuit of perfection is positively correlated with motive for success, which is related to the adaptation mode of achievement motivation [31,47]. Furthermore, a study mentioned that personal standard highly predicts the level of individual achievement motivation [48], implying that collegiate athletes with higher personal standard are likely to spend more time on training to pursue success. Moreover, different from previous studies, the result of current study showed a positive correlation between doubt about action and motive for success, which explained a small part of the variance of achievement motivation. A potential explanation may be that self-oriented perfectionists’ motive for success can influence their achievement goals and behaviors [49,50]. When athletes have doubts about their ability to complete tasks, which enable them to set a higher goal, resulting in extra training to achieve their goal.

For academic performance, we found that one aspect of perfectionistic concerns and two aspects of perfectionistic strivings were significantly associated with academic performance of collegiate athletes. Specifically, collegiate athletes with higher personal standard and organization levels showed better academic performance than those with lower. Previous studies have proven that the high standard dimension of positive perfectionism is positively correlated with students’ grades, and students with higher perfectionism beliefs tend to have good learning habits and perseverance, and work in a more organized way, which results in better academic performance [51]. This finding supports Hamachek’s theory that students with higher standards of positive perfectionism are more likely to achieve good grades [52]. Different from previous studies, this study found that collegiate athletes with higher level of doubt about action had better academic performance. A possible explanation is that when collegiate athletes doubt themselves to complete certain task, they may show an indomitable spirit to work hard in order to achieve their goal [53], resulting in better academic results. Furthermore, the result reported that the motivation to pursue success is positively correlated with academic performance and has a large effect size. According to Atkinson’s theory of achievement motivation, achievement motivation involves the emotional conflict between the expectation of success and the fear of failure [54], and the difference of achievement motivation is an important factor affecting the academic performance of students [55], showing that students with higher achievement motivation have better academic performance than those with lower [56,57]. Additionally, collegiate students with higher academic performance have higher motivation to pursue success [58,59], which is consistent with the results of current study.

In addition, the current research showed that there was a significant positive correlation between personal standard, doubt about action, and motive for success. Motive for success significantly mediated the relationship between personal standard and academic performance, as well as the relationship between doubt about action and academic performance. It is easy to explain that collegiate athletes who have higher standards tend to have a higher level of motivation to pursue success, resulting in achieve better academic results. On the other hand, collegiate basketball players who doubt themselves may spend more time on their study or training to achieve their goal [59].

From the perspective of psychology, extra training and academic performance are very important for collegiate athletes. However, there is limited research focusing on the relationship of specific aspects of perfectionism with extra training, academic performance, and the intermediary role of the achievement motivation in this relationship. This study can theoretically improve the basic research on perfectionism and achievement motivation. In terms of practical contribution, we suggest that coaches and teachers need to focus more on developing collegiate athletes’ success experience, improving their motivation for success, and designing appropriate lectures and training to improve their interest in learning, which may improve collegiate athletes’ personalities and motivate their success motivation. As a result, it may improve collegiate athletes’ sports performance through extra training, and devoting more time to pursue better academic performance.

Several limitations should be noted. Firstly, as the sampling was not based on a random approach, the participants may vary in relation to the actual patterns of the general collegiate athletes (e.g., in other individual or team sports, in female samples and in those who are majored in arts-related subjects). Therefore, the representativeness and generalizability of our findings should be further examined in future studies. Moreover, the extra-training was measured by self-reported items which might result in the recall bias and social desirability effects. Finally, although prospective design allowed for conclusion about the predictive validity of the mediation model, the causal relationship could not be confirmed and should be further examined by more strict experimental designs (e.g., randomized controlled trial).

## 5. Conclusions

This study is the first to examine the relationship between five aspects of perfectionism and extra training and academic performance as well as the mediating effect of achievement motivation among Chinese collegiate athletes. We found that only certain aspects of perfectionistic concerns and perfectionistic strivings should significant association with extra training and academic performance in the study samples. In addition, motive for success was identified as a salient mediator in the relationship between certain perfectionism aspects and extra training and academic performance. The current study provides a new perspective on the psychological mechanism of perfectionism and achievement motivation and provides the direction for future collegiate athletes to conduct extra training and improve their academic performance.

## Figures and Tables

**Figure 1 ijerph-19-10764-f001:**
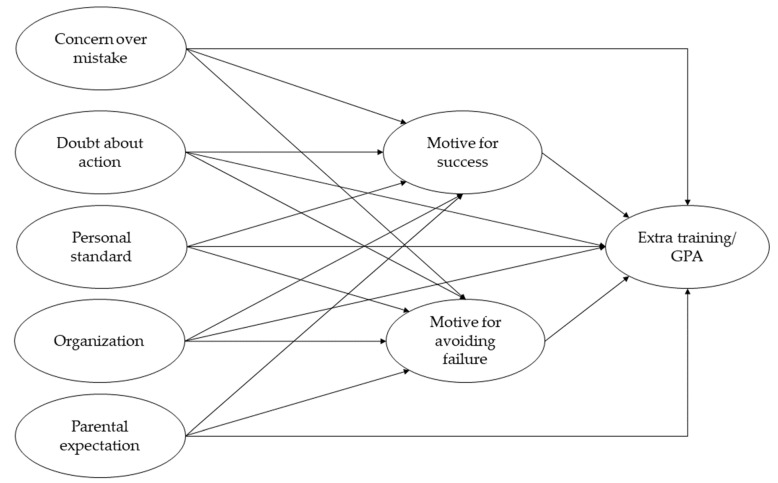
Hypothesized model of the mediating role of achievement motivation in the association of perfectionism with extra training and academic performance (GPA).

**Figure 2 ijerph-19-10764-f002:**
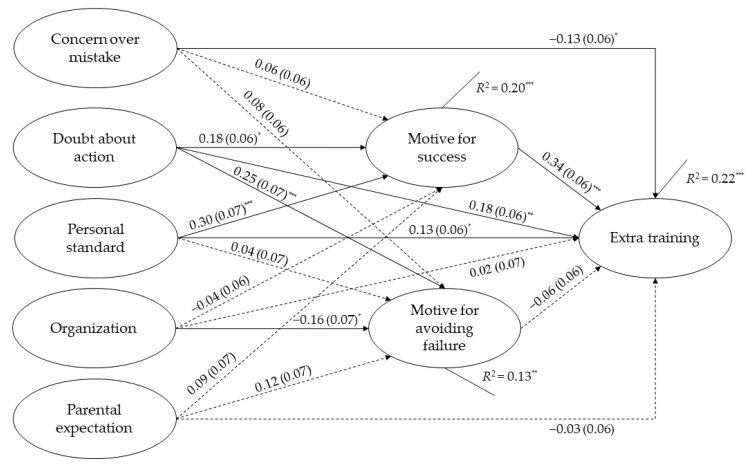
Final structural model with standardized path coefficients and standard errors for perfectionism, achievement motivation, and extra training (*n* = 243). All the demographics were included as covariates. Significant path is indicated by solid line and non-significant path is indicated by dotted line. * *p* < 0.05, ** *p* < 0.01, *** *p* < 0.001.

**Figure 3 ijerph-19-10764-f003:**
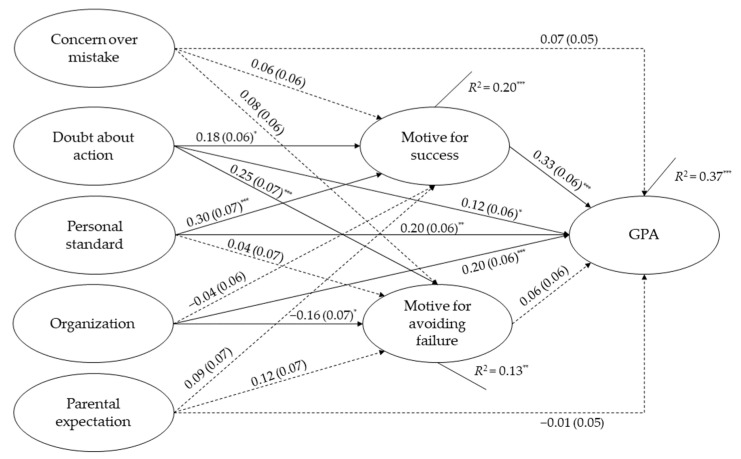
Final structural model with standardized path coefficients and standard errors for perfectionism, achievement motivation, and academic performance (GPA) (*n* = 243). All the demographics were included as covariates. Significant path is indicated by solid line and non-significant path is indicated by dotted line. * *p* < 0.05, ** *p* < 0.01, *** *p* < 0.001.

**Table 1 ijerph-19-10764-t001:** Model fit of the measurement model (*n* = 243).

Models	*α*	*χ* ^2^	*p*	*df*	χ^2^/*df*	CFI	TLI	RMSEA	90% CI of RMSEA	SRMR	Factor Loading
CM	0.863	22.555	0.02	11	2.050	0.980	0.963	0.066	0.025–0.104	0.031	0.379–0.831
PS	0.702	15.192	0.17	11	1.381	0.977	0.955	0.040	0.000–0.083	0.037	0.386–0.721
DA	0.739	Saturated measurement model									0.598–0.805
OR	0.725	13.505	0.14	9	1.501	0.969	0.948	0.045	0.000–0.092	0.038	0.425–0.740
PE	0.782	7.851	0.09	4	1.963	0.986	0.966	0.063	0.000–0.128	0.024	0.522–0.757
MS	0.884	76.497	<0.001	32	2.391	0.942	0.919	0.076	0.054–0.098	0.056	0.460–0.855
MF	0.873	65.637	<0.001	34	1.931	0.948	0.932	0.062	0.039–0.084	0.049	0.537–0.823

*χ*^2^ = Chi-square; *df* = degrees of freedom; CFI = comparative fit index; TLI = Tucker–Lewis index; RMSEA = root mean square error of approximation; CM, PS, DA, OR, PE denoted concern over mistake, personal standard, doubts about action, organization, and parental expectations, respectively; MS and MF denoted motive for success and motive for avoiding failure, respectively.

**Table 2 ijerph-19-10764-t002:** Means, standard deviations, ranges, and inter-correlations of the study variables (*n* = 243).

Variables	1	2	3	4	5	6	7	8	9
1. CM	1								
2. PS	0.09	1							
3. DA	0.29 **	0.29 **	1						
4. OR	0.03	0.39 **	0.09	1					
5. PE	0.24 **	0.41 **	0.32 **	0.30 **	1				
6. MS	0.16 **	0.39 **	0.31 **	0.14 *	0.30 **	1			
7. MF	0.18 **	0.11	0.31 **	−0.08	0.19 **	0.31 **	1		
8. ET	−0.03	0.31 **	0.26 **	0.15 *	0.18 **	0.40 **	0.08	1	
9. AP	0.18 *	0.44 **	0.33 **	0.32 **	0.28 **	0.49 **	0.22 **	0.22 *	1
Mean (SD)	3.01 (0.93)	3.51 (0.56)	3.12 (0.80)	3.98 (0.54)	3.11 (0.80)	2.51 (0.57)	2.25 (0.58)	3.26 (2.44)	3.29 (0.47)
Range	1–5	2.14–5	1–5	1.83–5	1–5	1–4	1–4	0–8	2–4

CM = concern over mistake; PS = personal standard; DA = doubts about action; OR = organization; PE = parental expectation; MS = motive for success; MF = motive for avoiding failure; ET = extra training; AP = academic performance; SD = standard deviation; * *p* < 0.05; ** *p* < 0.01.

**Table 3 ijerph-19-10764-t003:** Standardized parameter estimates for the direct, indirect, and total effects in the mediation models of extra training and academic performance in study samples (*n* = 243).

Effects	Extra Training (ET)				Academic Performance (GPA)			
	*β*	*p*	95%CI		*β*	*p*	95%CI	
			LB	UB			LB	UB
**Direct effects**								
CM→ET	−0.13	0.021	−0.22	−0.03				
PS→ET	0.13	0.045	0.02	0.24				
DA→ET	0.18	0.005	0.07	0.28				
OR→ET	0.02	0.71	−0.09	0.13				
PE→ET	−0.03	0.63	−0.13	0.07				
MS→ET	0.34	<0.001	0.23	0.43				
MF→ET	−0.06	0.26	−0.15	0.03				
CM→GPA					0.07	0.15	−0.01	0.14
PS→GPA					0.20	0.001	0.10	0.29
DA→GPA					0.12	0.037	0.02	0.21
OR→GPA					0.20	<0.001	0.11	0.29
PE→GPA					−0.01	0.82	−0.09	0.07
MS→GPA					0.33	<0.001	0.23	0.43
MF→GPA					0.06	0.30	−0.03	0.15
**Indirect effects**								
CM→MS→ET	0.02	0.38	−0.02	0.05				
PS→MS→ET	0.10	0.002	0.06	0.17				
DA→MS→ET	0.06	0.027	0.02	0.11				
OR→MS→ET	−0.01	0.58	−0.05	0.02				
PE→MS→ET	0.03	0.23	−0.01	0.08				
CM→MF→ET	−0.01	0.43	−0.02	0.001				
PS→MF→ET	−0.003	0.69	−0.02	0.003				
DA→MF→ET	−0.02	0.30	−0.05	0.004				
OR→MF→ET	0.01	0.35	−0.002	0.03				
PE→MF→ET	−0.01	0.38	−0.03	0.001				
CM→MS→GPA					0.02	0.37	−0.01	0.05
PS→MS→GPA					0.10	0.003	0.05	0.16
DA→MS→GPA					0.06	0.037	0.02	0.11
OR→MS→GPA					−0.01	0.58	−0.05	0.02
PE→MS→GPA					0.03	0.21	−0.01	0.08
CM→MF→GPA					0.01	0.49	−0.001	0.02
PS→MF→GPA					0.002	0.74	−0.004	0.02
DA→MF→GPA					0.02	0.34	−0.01	0.05
OR→MF→GPA					−0.01	0.42	−0.03	0.003
PE→MF→GPA					0.01	0.41	−0.002	0.03
**Total effects**								
CM→ET	−0.12	0.046	−0.22	−0.02				
PS→ET	0.23	<0.001	0.12	0.33				
DA→ET	0.22	0.001	0.11	0.33				
OR→ET	0.02	0.75	−0.09	0.13				
PE→ET	−0.004	0.94	−0.10	0.10				
CM→GPA					0.09	0.08	0.01	0.17
PS→GPA					0.30	<0.001	0.19	0.39
DA→GPA					0.19	0.001	0.10	0.28
OR→GPA					0.18	0.003	0.08	0.28
PE→GPA					0.03	0.64	−0.07	0.13

*β* = Standardized parameter estimate; 95% CI = 95% confidence interval of standardized parameter estimate; LB = lower bound of 95% CI; UB = upper bound of 95% CI; CM = concern over mistake; PS = personal standard; DA = doubts about action; OR = organization; PE = parental expectation; MS = motive for success; MF = motive for avoiding failure; →: indicating the former variable predicts the latter one. All demographics were included as covariates.

## Data Availability

Data are available by contacting the corresponding authors.

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
