# Peer review of "Relationships between Perfectionism, Extra Training and Academic Performance in Chinese Collegiate Athletes: Mediating Role of Achievement Motivation"

_ijerph, 2022, doi:10.3390/ijerph191710764_

Round 1

Reviewer 1 Report

The study itself is extremely interesting and addresses important issues in both education and  sports.

There are certain phrases in the introduction, however, that may mislead the reader. For example, Line 32 – the word "trait" must be removed. Personality is a complex of traits and not a singular trait, as the authors further explain.

The operationalization of perfectionism is unclear. There is good perfectionism, when the individual pays attention to the details of their work and is performance-oriented, and bad perfectionism, when the individual gets too lost in the details, misses deadlines, and becomes anxious. So, we suggest delimiting the two types of perfectionism. Which of them does the study refer to? What should we expect? A positive or a negative association of perfectionism with the rest of the variables? All this information about perfectionism is found in the introduction, but it is presented chaotically. We recommend placing them coherently in the text.

Line 103  and the following lines – assumptions should include the direction of the relationship, not just its nature. In which direction is perfectionism associated with achievement motivation and with extra training? Positive or negative? In what sense does mediation occur? Does it ameliorate or accentuate the relationship?

Line 163 – perhaps the first part of the sentence should be reworded ("30 items regarding this scale" – does not sound very English).

Line 168 - "the psychology of achievement motivation" - maybe it would be better to remove "psychology of".

Line 172 – where the 0-4 GPA scores come from should be explained; how is grading done in China? What does score 0 mean? That the athlete has a very low grade or no grade at all?

Line 218 and the following lines – the description of the results shows both positive and negative relationships. Their meaning must be highlighted in words.

Indeed, in the discussion part it is mentioned whether the relationships are negative or positive, but we consider it important to know from the beginning the expectations of the authors in accordance with the literature presented in the introduction for the construction of assumptions.

Author Response

Reviewer 1:

The study itself is extremely interesting and addresses important issues in both education and sports.

[Response] We are grateful for your valuable comments and positive feedback!

There are certain phrases in the introduction, however, that may mislead the reader. For example, Line 32 – the word "trait" must be removed. Personality is a complex of traits and not a singular trait, as the authors further explain.

[Response] Thank you for pointing out this issue. We have deleted the word “trait” accordingly.

The operationalization of perfectionism is unclear. There is good perfectionism, when the individual pays attention to the details of their work and is performance-oriented, and bad perfectionism, when the individual gets too lost in the details, misses deadlines, and becomes anxious. So, we suggest delimiting the two types of perfectionism. Which of them does the study refer to? What should we expect? A positive or a negative association of perfectionism with the rest of the variables? All this information about perfectionism is found in the introduction, but it is presented chaotically. We recommend placing them coherently in the text.

[Response] Thank you for your constructive suggestions. We have reworked on the introduction part accordingly. We focused on both good and bad perfectionism, namely perfectionistic strivings, and perfectionistic concerns. In our study, we examined five specific facets of perfectionism, where concern over mistake and doubt about action were of perfectionistic concerns, and personal standard, organization, and parental expectation were of perfectionistic strivings. We have also elaborated more on the positive and negative associations of perfectionism with the rest of the variables and enhanced the overall logic of the background knowledge.

Line 103 and the following lines – assumptions should include the direction of the relationship, not just its nature. In which direction is perfectionism associated with achievement motivation and with extra training? Positive or negative? In what sense does mediation occur? Does it ameliorate or accentuate the relationship?

[Response] Thank you for your valuable comments. We have reworked on the hypotheses part accordingly.

Line 163 – perhaps the first part of the sentence should be reworded ("30 items regarding this scale" – does not sound very English).

[Response] Thank you for your suggestion. We have rephrased this part accordingly. It now reads like: “This scale consists of 30 items, which are divided into 2 subscales: motive for success dimension and motive for avoiding failure.” 

Line 168 - "the psychology of achievement motivation" - maybe it would be better to remove "psychology of".

[Response] Thank you for your suggestion. We have rephrased this part accordingly. It now reads like: “A higher total score of the scale (i.e., the score of the motivation for success subscale minus the score of the motivation for avoiding failure subscale) indicates a stronger achievement motivation.” 

Line 172 – where the 0-4 GPA scores come from should be explained; how is grading done in China? What does score 0 mean? That the athlete has a very low grade or no grade at all?

[Response] Thank you for your questions. We have elaborated more on this point. It now reads like: “The semester GPA was collected directly from the university records, which was calculated using the credit-weighted sum of the grades for all courses divided by the total credits. The GPA was coded on a continuous scale ranging from “A=4” to “F=0 (failed)”” 

Line 218 and the following lines – the description of the results shows both positive and negative relationships. Their meaning must be highlighted in words.

[Response] Thank you for your suggestions. We have revised this part accordingly.

Indeed, in the discussion part it is mentioned whether the relationships are negative or positive, but we consider it important to know from the beginning the expectations of the authors in accordance with the literature presented in the introduction for the construction of assumptions.

[Response] Thank you for your valuable suggestions. We have improved the discussion part accordingly.

Reviewer 2 Report

The current study investigated a very interesting issue. 

The study has been rigorously designed and conducted, with the use of more complex statistical approaches as mediation analysis, which increases the value of the study with the aim to evaluate association among variables. 

The manuscript is clearly presented with a high quality of scientific writing style. 

I would like to make sure about results section and figures. I see some incongruence among data in the text and figures. I suggest authors to make a double check. Moreover, could authors explain in figure 2 and 3 the data in bracket? What those data refer to?

Line 337-339. Authors are invited to implement the practical implications section since a general sentence has been provided regarding the improvement of players personality and motivation. How can they improve? Please expand and provide practical example. 

Author Response

Reviewer 2:

The current study investigated a very interesting issue. 

The study has been rigorously designed and conducted, with the use of more complex statistical approaches as mediation analysis, which increases the value of the study with the aim to evaluate association among variables. The manuscript is clearly presented with a high quality of scientific writing style.

[Response] We are grateful for your valuable comments and positive feedback! 

I would like to make sure about results section and figures. I see some incongruence among data in the text and figures. I suggest authors to make a double check. Moreover, could authors explain in figure 2 and 3 the data in bracket? What those data refer to?

[Response] Thank you for your constructive comments. We have made a double check and found that the text and figures are consistent. In addition, the data in bracket refers to the standard error. Based on your suggestion, we have added this information in the text as well as in the caption of the figures.

Line 337-339. Authors are invited to implement the practical implications section since a general sentence has been provided regarding the improvement of players personality and motivation. How can they improve? Please expand and provide practical example. 

[Response] Thank you for your valuable suggestions. We have elaborated more on this point. It now reads like:

"From the perspective of psychology, the extra training and academic performance are very important for collegiate athletes. However, there is limited research focusing on the relationship of specific aspects of perfectionism with extra training, academic performance, and the intermediary role of the achievement motivation in this relationship. This study can theoretically improve the basic research on perfectionism and achievement motivation. In terms of practical contribution, we suggest that coaches and teachers need to focus more on developing collegiate athletes' success experience, improving their motivation for success, and designing appropriate lectures and training to improve their interest in learning, which may improve collegiate athletes' personalities and motivate their success motivation. As a result, it may improve collegiate athletes' sports performance through extra training, and devoting more time to pursue better academic performance."

Reviewer 3 Report

Dear Authors

sincerely

Author Response

Reviewer 3:

Dear Authors

This was a very interesting study of college basketball players analyzed using SEM (mediation model) to examine whether achievement motivation plays a mediating role in the relationship between perfectionism, special training, and academic performance. In reviewing the article, several points caught my attention, which are listed below. Please respond to them.

[Response] Thank you for your time and valuable comments.

Abstract

  1. Please describe in such as “(Abbreviation, Formal name; Abbreviation, Formal name; ...)”.

[Response] Thank you for your suggestion. We described it as full spelling followed by the abbreviation, because this is the journal format requirement. 

Introduction

  1. Although you have focused on basketball players, have you conducted similar surveys of other athletes? If so, I would appreciate a review of the results. In particular, it would be good if the reasons for limiting it to basketball players are also included there.

[Response] Thank you for your insightful suggestion. With a convenience sampling, we only selected basketball players and did not conduct similar surveys in other athletes. This should be a limitation and we have added this in the discussion part.

"Firstly, as the sampling was not based on a random approach, the participants may vary in relation to the actual patterns of the general collegiate athletes (e.g., in other individual or team sports, in female samples and in those who are majored in arts-related subjects). Therefore, the representativeness and generalizability of our findings should be further examined in future studies. "

  1. Are there any reports of direct effects other than indirect effects that have been investigated in basketball players? It would be better to review this as well and strengthen the reasons for conducting the analysis in a mediation analysis.

[Response] Thank you for your valuable comments. We have added more literature and substantially revised the introduction part.

 Materials and Methods

  1. Among the study participants, there are large differences between men and women, but I suspect that this may have a significant impact on the results of the study. What are your thoughts on this point? (L 133)

[Response] Thank you for your constructive suggestions. Actually, in our model analysis, we had included all the demographics as covariates. We have added this information in the statistical analysis and footnotes of figures and tables. In addition, we agree that this point may have a significant impact on the findings in terms of the representativeness and generalizability. We have highlighted this point in the limitation part accordingly.

  1. Normally distributed -> parametric distributed
  2. You wrote it followed a parametric distribution; how did you confirm this? (Line 177)

[Response] For Point 2 & 3, thank you for your comments. In our study, we used the Q-Q plot and S-K test to detect the data distribution. As the maximum likelihood estimation was used for the model analysis in our study, whereby the standard errors and tests of model fit were robust with respect to the observed variables with non-normal and non-independent distribution. We have rephrased the description of statistical analysis to avoid confusion.

Results

  1. In Figure 2 and 3, please describe in the text and in footnotes what is represented by the dotted lines. Likewise, please also state what the bolded words refer to.

[Response] Thank you for your valuable comments. We have updated the figures and added this information in the footnotes accordingly.

Reviewer 4 Report

Thank you very much for the opportunity to read the research paper. The following issues are suggested to be considered:

The discussion chapter might be improved by showing a debate between this research results and more recent research publications. This might require also the expansion of a literature review so that a research gap would be clearer and based on more recent research.

More careful editing is suggested. All the figures and tables should be explained in the text. E.g. there is Figure 3 which is not in the text and Figure 4 is in the text but there is no Figure 4 presented.

Figure 1 requires to be more explained in relation to hypotheses.

Conclusions might be more focused as the results show a lot more than is presented in the conclusions chapter.

Research ethics is an important issue in this paper. The authors in the 129 line present that no severe mental and cognitive disorders as one of the criteria to participate in the research. Though this is an issue of very sensitive confidential personal data and is not a typical data collection which can be covered by signing a form. The research ethics including data collection, processing and archiving is required to be explained in the publication.

Author Response

Reviewer 4:

Thank you very much for the opportunity to read the research paper. The following issues are suggested to be considered:

The discussion chapter might be improved by showing a debate between this research results and more recent research publications. This might require also the expansion of a literature review so that a research gap would be clearer and based on more recent research.

[Response] Thank you for your constructive suggestions. We have improved the discussion accordingly.

More careful editing is suggested. All the figures and tables should be explained in the text. E.g. there is Figure 3 which is not in the text and Figure 4 is in the text but there is no Figure 4 presented.

[Response] Thank you for pointing out this typo. There are three figures in our manuscript, and we have changed Figure 4 to Figure 3.

Figure 1 requires to be more explained in relation to hypotheses.

[Response] Thank you for your constructive comments. We have revised the study hypotheses accordingly.

Conclusions might be more focused as the results show a lot more than is presented in the conclusions chapter.

[Response] Thank you for your suggestions. We have revised the conclusion part accordingly.

Research ethics is an important issue in this paper. The authors in the Line 129 present that no severe mental and cognitive disorders as one of the criteria to participate in the research. Though this is an issue of very sensitive confidential personal data and is not a typical data collection which can be covered by signing a form. The research ethics including data collection, processing and archiving is required to be explained in the publication.

[Response] Thank you for highlighting this issue. We agree that the research ethics is very important. Actually, in our study, the study has been approved by the research committee of our university and all participants have signed the written informed consent form. For the screening of mental and cognitive disorders, this is overlapped with being a member of university sports team, as when the university sports team selected the eligible team members, the person who has severe mental and cognitive disorders (identified by the professional measurement tools) was excluded due to the safety concerns. Based on your suggestions, we have deleted this point and revised this part accoordingly. 

Reviewer 5 Report

The Monte Carlo simulation method is not properly described; even the normalization of the variables is not mentioned.

The references should be carefully reviewed, there are inconsistencies between them and the citations in the text. For example:

35. Hagtvet, J. Psychological Development and Education 1992, 2, 14-16.

Renmin, Y. & Hagtvet, K. J. Psychological Development and Education 1992, 2, 14-16.

34. Zi Fei. Psychological science 2004, 27,943-945.

The reference of 34. Zi Fei. Psychological science 2004, 27,943-945. wasn`t available, instead of the found source was The Development of the Zi`s Positive Perfectionism Scale (ZPPS) Fei Zi

Author(s): Fei Zi

Pp: 27-33 (7)

Doi: 10.2174/978160805186111001010027

In the method section it is mentioned that:

The model fit was evaluated 181 

using several goodness-of-fit indices, including Chi-square (X2), Chi squared/deviation 182,

However, in the fits of the two models presented, and of the different scales and subscales, this fit value is not taken into account.

It is not clear why the authors do not present the standardized coefficients in terms of the effect size for each of the paths in the models (it is inferred that the values in parentheses are the error values)?

In addition, the quality of the figures is very poor, and the differences between the thickness and continuity of the lines representing the paths are not explained (apparently, thick and continuous lines correspond to significant coefficients; while dashed and light lines represent non-significant coefficients within the models).

Author Response

Reviewer 5:

The Monte Carlo simulation method is not properly described; even the normalization of the variables is not mentioned.

[Response] Thank you for your comments. Actually, in our study, we used two methods to estimate the sample size. One is the common rule of thumb for SEM (10 times the number of variables) and another is that based on the suggestion of the Fan et al., (1992) the sample size of 200 could achieve a moderate-to-large effect estimate. However, to avoid confusing readers, we have revised the sample size estimate part accordingly.

In addition, for the normalization of variables, as we employed the Mplus to conduct the SEM path analysis for mediation analyses rather than the step-by-step regressions and there were no interaction items in the analysis, there is no restriction for normalizing the variables and the standardized coefficients could be automatically calculated by the software. Therefore, we did not normalize the variables and this point was not highlighted in the manuscript.

The references should be carefully reviewed, there are inconsistencies between them and the citations in the text. For example:

[Response] Thank you for your suggestions. We have made a double check and corrected the inconsistencies in references.

  1. Hagtvet, J. Psychological Development and Education 1992, 2, 14-16.

Renmin, Y. & Hagtvet, K. J. Psychological Development and Education 1992, 2, 14-16.

  1. Zi Fei. Psychological science 2004, 27,943-945.

The reference of 34. Zi Fei. Psychological science 2004, 27,943-945. wasn`t available, instead of the found source was The Development of the Zi`s Positive Perfectionism Scale (ZPPS) Fei Zi

Author(s): Fei Zi

Pp: 27-33 (7)

Doi: 10.2174/978160805186111001010027

In the method section it is mentioned that:

The model fit was evaluated 181 using several goodness-of-fit indices, including Chi-square (X2), Chi squared/deviation 182, However, in the fits of the two models presented, and of the different scales and subscales, this fit value is not taken into account.

[Response] Thank you for your comments. As indicated in our original manuscript, “The general criteria for an acceptable model fit were <5 for c2/df, >0.90 for CFI and TLI, and < 0.08 for RMSEA and SRMR [36-37] (lines)”, we examined the fits of two models based on these criteria. We had reported the model fit of the measurement model in Table 1 and the model fit of mediation model in the first paragraph of “Main Analysis” section.

It is not clear why the authors do not present the standardized coefficients in terms of the effect size for each of the paths in the models (it is inferred that the values in parentheses are the error values)?

[Response] Thank you for your questions. For the effect size of the whole mediation model, we used the Cohen’s f2 which was calculated by using the equation “f2=R2/(1-R2)”. However, as we used the SEM path analysis rather than step-by-step regressions to examine the mediation effects, it is not applicable to estimate the R2 for each single path. Therefore, we did not report the Cohen’s f2 for each single path. Importantly, we reported the standardized effect coefficient β and standard errors or 95% CI, which is a common practice in psychology and behavioral sciences domains. The standardized β itself can clearly indicate the strength of the effect. In addition, the data in the bracket refers to the standard error. We have added this information in the footnotes of figures accordingly.

In addition, the quality of the figures is very poor, and the differences between the thickness and continuity of the lines representing the paths are not explained (apparently, thick and continuous lines correspond to significant coefficients; while dashed and light lines represent non-significant coefficients within the models).

[Response] Thank you for your valuable comments. We have updated all the figures and added more information in the footnotes.

Round 2

Reviewer 4 Report

The authors have made amendments and commented on the suggestions of the reviewer.